# High-Throughput Analysis of Neutrophil Extracellular Trap Levels in Subtypes of People with Type 1 Diabetes

**DOI:** 10.3390/biology12060882

**Published:** 2023-06-19

**Authors:** Samal Bissenova, Mijke Buitinga, Markus Boesch, Hannelie Korf, Kristina Casteels, An Teunkens, Chantal Mathieu, Conny Gysemans

**Affiliations:** 1Clinical and Experimental Endocrinology, Department of Chronic Diseases and Metabolism (CHROMETA), KU Leuven, 3000 Leuven, Belgium; samal.bissenova@kuleuven.be (S.B.); chantal.mathieu@uzleuven.be (C.M.); 2Department of Nutrition and Movement Sciences, Maastricht University, 6211 LK Maastricht, The Netherlands; m.buitinga@maastrichtuniversity.nl; 3Department of Radiology and Nuclear Medicine, Maastricht University Medical Centre, 6211 LK Maastricht, The Netherlands; 4Laboratory of Hepatology, Department of Chronic Diseases and Metabolism (CHROMETA), KU Leuven, 3000 Leuven, Belgium; markus.boesch@kuleuven.be (M.B.); hannelie.korf@kuleuven.be (H.K.); 5Woman and Child, Department of Development and Regeneration, KU Leuven, 3000 Leuven, Belgium; kristina.casteels@kuleuven.be; 6Anesthesiology and Algology, Department of Cardiovascular Sciences, KU Leuven, 3000 Leuven, Belgium; an.teunkens@uzleuven.be

**Keywords:** neutrophils, neutrophil extracellular trap (NET), type 1 diabetes, autoimmunity

## Abstract

**Simple Summary:**

The involvement of the innate immune system in autoimmune diseases, such as type 1 diabetes, is becoming more and more apparent. Perhaps the most overlooked innate immune cell type, the neutrophil, is one of the culprits for hyperactive immune responses (against oneself) in these diseases. However, the heterogeneous nature, both within and between these diseases, and the lack of standardized methods to study neutrophil function has hampered advancement in this research area. Our study used a live-cell imaging technique that allows for an unbiased and automated analysis of neutrophil function in children and adults with type 1 diabetes. Overall, neutrophils of people at different developmental stages of type 1 diabetes, irrespective of age, behaved similarly to those of healthy donors, despite some minor changes in peripheral neutrophil measures.

**Abstract:**

Neutrophils might play an important role in the pathogenesis of autoimmune diseases, including type 1 diabetes (T1D), by contributing to immune dysregulation via a highly inflammatory program called neutrophil extracellular trap (NET) formation or NETosis, involving the extrusion of chromatin entangled with anti-microbial proteins. However, numerous studies reported contradictory data on NET formation in T1D. This might in part be due to the inherent heterogeneity of the disease and the influence of the disease developmental stage on neutrophil behavior. Moreover, there is a lack of a standardized method to measure NETosis in an unbiased and robust manner. In this study, we employed the Incucyte^®^ ZOOM live-cell imaging platform to study NETosis levels in various subtypes of adult and pediatric T1D donors compared to healthy controls (HC) at baseline and in response to phorbol–myristate acetate (PMA) and ionomycin. Firstly, we determined that the technique allows for an operator-independent and automated quantification of NET formation across multiple time points, which showed that PMA and ionomycin induced NETosis with distinct kinetic characteristics, confirmed by high-resolution microscopy. NETosis levels also showed a clear dose-response curve to increasing concentrations of both stimuli. Overall, using Incucyte^®^ ZOOM, no aberrant NET formation was observed over time in the different subtypes of T1D populations, irrespective of age, compared to HC. These data were corroborated by the levels of peripheral NET markers in all study participants. The current study showed that live-cell imaging allows for a robust and unbiased analysis and quantification of NET formation in real-time. Peripheral neutrophil measures should be complemented with dynamic quantification of NETing neutrophils to make robust conclusions on NET formation in health and disease.

## 1. Introduction

Neutrophils have increasingly gained attention in the past decade for their prominent role in pathological contexts, most notably in autoimmune diseases such as type 1 diabetes (T1D) [1,2]. Characterized by an immune-mediated attack on the insulin-producing beta cells of the pancreas, T1D is a debilitating chronic disease with a growing global incidence, particularly in young children [3,4]. Although neutrophils were detected among pancreas-infiltrating immune cells, before and at disease onset, their role in T1D remains unclear [5]. Elucidating the implications of neutrophils and their functions in T1D development and progression could potentially lead to novel therapeutic targets.

Abundant in circulation, neutrophils are responsible for patrolling the organism for potential pathogens and, when called upon, are the first to infiltrate the site of inflammation or infection [6]. They are multifunctional cells that can employ functions such as degranulation, reactive oxygen species (ROS) production and release, phagocytosis, and most notably, neutrophil extracellular trap (NET) formation or NETosis [6]. The latter is characterized by the release of DNA entangled with anti-microbial proteins such as myeloperoxidase (MPO), proteinase 3 (PR3), and neutrophil elastase (NE) in a mesh-like structure [7]. Different stimuli induce different pathways of NET formation that employ various neutrophil components for histone cleavage necessary for nuclear decondensation. Various types of bacteria, as well as phorbol–myristate acetate (PMA) depend on ROS production mediated by nicotinamide adenine dinucleotide phosphate (NADPH) oxidase (NOX) and on MPO and NE for histone cleavage [8,9]. On the other hand, calcium-ionophores such as ionomycin and nigericin induce NETs independently of NOX and rely on mitochondrial ROS and peptidyl arginine deiminase 4 (PAD4) for histone cleavage [8,10]. NETosis plays an essential role in pathogen elimination, but when uncontrolled, it can cause hyper-inflammation and substantial damage to the surrounding tissues [11]. In fact, dysregulated NETosis has been shown to play a key role in autoimmune diseases such as rheumatoid arthritis (RA), systemic lupus erythematosus (SLE), and anti-neutrophil cytoplasmic antibody (ANCA)-associated vasculitis (AAV) [12,13,14,15,16,17]. Their role in T1D remains to be elucidated.

Although a significant proportion of the pancreas-infiltrating neutrophils were shown to undergo NET formation, questions concerning the levels of NETosis remain [5]. Moreover, studies reported contradictory evidence on peripheral levels of NET remnants or NETosis in people at different T1D stages. In adult autoantibody-positive and newly diagnosed T1D individuals, serum levels of MPO and NE were shown to be increased, suggesting enhanced NETosis [18]. This was also the case in the pediatric population, where increased levels of MPO and NE, as well as other NET-associated biomarkers such as PR3, PAD4, cell-free DNA-histone complexes, and extracellular DNA were observed at the time of T1D onset [19,20]. However, other studies have reported lower MPO-DNA markers as well as reduced NE and PR3 levels in adults with established T1D compared to healthy control (HC) donors [21,22]. The heterogeneity of NETosis levels in T1D could be explained by the influence of different disease stages on NET formation and the lack of a standardized technique to measure NETosis.

T1D is not only a heterogeneous disease but it is also characterized by distinct stages of disease progression [23]. These stages are characterized by the appearance of circulating autoantibodies indicative of islet autoimmunity (stage 1), the onset of glucose intolerance by ensuing beta-cell destruction (stage 2), eventually culminating in clinical symptoms such as polyuria, polydipsia, weight loss, fatigue, hyperglycemia, and sometimes diabetic ketoacidosis (stage 3) [24]. Additionally, the emerging concept of neutrophil heterogeneity points to the potential influence of micro-environmental cues, particular to disease stage, on neutrophil behavior and functions, such as NETosis [25]. As such, the various T1D stages might have an impact on the reported NETosis levels in the aforementioned studies. As for the techniques used to determine levels of NETosis, the studies described either indirect measurements of NET formation (i.e., serum or plasma levels of NET markers such as MPO and NE or extracellular DNA) or employed error-prone, laborious, and relatively subjective methods to measure NETosis levels (i.e., manual counting) [18,19,20,21,22]. It could be argued that these discrepancies might affect the reported NETosis levels. In fact, as NET formation gains more attention in pathological contexts, there is growing need for a more standardized technique for analyzing and measuring the process.

Therefore, in this study, we determined the levels of NETosis in people at various stages of T1D development and HC, using Incucyte^®^ ZOOM, a high-throughput and robust live-cell imaging technique that offers significant advantages because many samples can be rapidly imaged using clearly defined NET parameters. Our findings highlight this technique, which uses a dual-dye protocol and automated quantification, as a robust and unbiased method to quantify NET formation in individuals at various stages of T1D development.

## 2. Materials and Methods

### 2.1. Human Subjects and Ethics Statement

Peripheral blood was collected by venipuncture between the hours of 8:00 and 11:00 from T1D and HC donors recruited at the University hospital UZ Leuven. Subjects signed informed consent; all experiments involving human subjects were approved by the Ethics Committee Research UZ/KU Leuven. Demographics for adult and pediatric T1D and HC donors in the study are shown in Table 1 and Table 2. All new-onset and established T1D donors had active disease as determined by a clinician at the University hospital UZ Leuven and were undergoing insulin replacement therapy. Lymphocyte, monocyte, and granulocyte counts were determined for each donor using ABX automated hematology analyzer (Horiba, Kyoto, Japan). Autoantibodies against glutamic acid decarboxylase (GAD; GADA), insulinoma-associated protein (IA-2; IA-2A), insulin (IAA), and zinc transporter 8A (ZnT8A) were determined in stored plasma by liquid-phase radio-binding assay (UZ Brussel, Brussels, Belgium).

### 2.2. Isolation of Primary Human Neutrophils

Peripheral blood samples from T1D and sex- and age-matched HC donors were collected in EDTA-coated tubes and processed within 30 min of sampling at room temperature under sterile conditions. Neutrophils were isolated as previously described [26]. Briefly, cell types were separated by density centrifugation (600× *g* for 30 min) using Lymphoprep gradient (Stem Cell, Cat No. 07851, Saint Égrève, France). Sediment containing neutrophils was separated from red blood cells using a 1% gelatine solution (Sigma, Cat No. G-9382, St. Louis, MO, USA) and residual red blood cells were removed by hypotonic lysis using MilliQ water. Plasma from each donor was stored at −80 °C within 30 min of sampling.

### 2.3. Measurements of NET Markers

Levels of MPO and NE in plasma samples were determined using the R-PLEX human myeloperoxidase antibody set (Meso Scale Discovery (MSD), Cat No. ab F214E-3, Gaithersburg, MD, USA) and human PMN elastase ELISA kit (Abcam, Cat No. ab119553, Cambridge, UK), respectively, following the manufacturer’s instructions. MSD and ELISA plates were analyzed on the MESO QuickPlex SQ120 (MSD) and the VICTOR™ (spectrophotometry, absorbance at 450 nm, Perkin Elmer, Zaventem, Belgium), respectively.

### 2.4. NET Induction and Staining for Incucyte^®^ ZOOM

Neutrophils were resuspended at 2 × 10^6^ cells/mL and incubated with NUCLEAR-ID^®^ red DNA stain (1 µL per 1.5 × 10^6^ cell suspension, Enzo Life Sciences, Farmingdale, NY, USA) for 5 min in the dark. Cells were washed 3 times in RPMI medium and spun down at 2400 g for 5 min. Stained cells were then plated at 20,000 cells per 100 µL in a 96-well poly-l-lysine coated flat bottom plate and incubated for 20 min to settle. Cells were then stimulated or not with PMA or ionomycin at given concentrations in a mixture with 2 µM SYTOX^®^ Green DNA dye (Thermo Fisher, Cat No. S7020, Waltham, MA, USA) and were placed in the Incucyte^®^ ZOOM platform contained within an incubator at 37 °C and 5% CO_2_.

### 2.5. Imaging and Quantification of NETs with Incucyte^®^ ZOOM

The cells were imaged using phase contrast, red (800-ms exposure) and green (300-ms exposure) channels of the Incucyte^®^ ZOOM live-cell imaging platform. Spectral unmixing of 14.5% of red removed from the green was applied according to manufacturer’s instructions. Two images from distinct regions in each well were taken every 20 min using the 10× or 20× objectives, and each condition was run in triplicate. We adapted the protocol used by Gupta et al. [27] to establish an optimal processing definition with parameters to determine imaged objects as live cells and NETing cells. We employed the Top-Hat method for background correction, the edge split tool to accurately distinguish between closely spaced objects and thresholds for fluorescence and area size. The processing definition was set to recognize live neutrophils as cells positive for the membrane-permeable NUCLEAR-ID^®^ red DNA stain (red channel) with a minimum area of 35 µM^2^ and maximum eccentricity (roundness) of 0.85. NETing neutrophils were recognized as cells positive for the membrane-impermeable SYTOX^®^ Green DNA dye (green channel) with a minimum area of 100 µM^2^ to account for the increase in nuclear diameter due to nuclear decondensation and minimum intensity of 5000 green corrected units (GCU) to exclude debris. Additionally, filters were applied to exclude objects below the radius of 10 µM and fluorescent thresholds of 0.5 red corrected units (RCU) and 1 GCU, for the red and green channels, respectively. The edge split tool was set to −10 for the green channel. Using these parameters of the processing definition, live and NET cell counts (1/mm^2^) were used to determine the percentage of NETosis relative to the total amount of cells. The details of the image processing steps and parameters of the processing definition, as well as representative images showing the application of the processing definition on imaged cells are summarized in Figure 1.

### 2.6. Visualization of NETs by Confocal Microscopy

Isolated neutrophils were seeded on 0.01% poly-l-lysine (Sigma, P4832, St. Louis, MO, USA) coated coverslips at 500,000 cells/mL in RPMI 1640 medium (Thermo Fischer, Cat No. 52400-025) supplemented with 10% fetal bovine serum (FBS) and incubated for 20 min at 37 °C in 5% CO_2_ for adherence. Cells were then stimulated with 10 nM of PMA (Sigma, Cat No. P8139) or 10 µM of ionomycin (Sigma, Cat No. I0634) for 3 h at 37 °C in 5% CO_2_. Coverslips were washed with warm PBS and cells were fixed with 4% paraformaldehyde (Klinipath/VWR international, Cat No. 4177; Amsterdam, The Netherlands) for 15 min. Cells were then washed and stained with 3 µM of SiR-DNA (Spirochrome, Cat No. SC007, Switzerland) for 30 min, washed with medium, and mounted on slides. Images were recorded on an Abberior STED Expert line (Cell and Tissue Imaging Cluster, CIC) microscope using the 63× and 100× objectives, for confocal and STED microscopy, respectively, supported by FWO I001918N to Pieter Vanden Berghe, University of Leuven. Images were deconvolved and processed using Huygens (Scientific Volume Imaging, Hilversum, The Netherlands) and ImageJ/Fiji software (Java; NIH, Bethesda, MD, USA).

### 2.7. Bioinformatics and Statistical Analyses

Statistical analysis was performed on Prism software 9 (GraphPad, La Jolla, CA, USA) using the unpaired Kruskal–Wallis test with Bonferroni correction. A *p*-value of <0.05 was considered statistically significant.

## 3. Results

### 3.1. Identification of NET Formation in Response to Distinct Stimuli by Incucyte^®^ ZOOM

Using Incucyte^®^ ZOOM, we were able to observe distinct morphological changes induced by stimulating neutrophils with PMA or ionomycin. While PMA-stimulated neutrophils were characterized by gradual nuclear swelling and increase in SYTOX^®^ green dye staining within 80 min of stimulation, ionomycin induced a delayed (at the 160-min time point), but rapid shift from the polylobal nuclear morphology to a decondensed and SYTOX^®^ green positive morphology (Figure 2A,B, Appendix A). The kinetic characteristics of each stimulus were confirmed by the distinct curves showing NET formation across multiple time points (Figure 2C,D). Furthermore, the cells showed a dose response to increasing concentrations of each stimulus, allowing us to focus on an optimal dose to study NET formation for subsequent analyses (Figure 2C,D).

Confocal and STED microscopy images of PMA- and ionomycin-stimulated neutrophils confirmed the formation of NETs in response to the stimuli (Figure 2E,F). They also showed that the two stimuli induced NETosis in a morphologically distinct manner. While PMA stimulation induced a morphology consistent with cell swelling (Figure 2E), ionomycin induced protrusions of long filaments of DNA (Figure 2F).

### 3.2. Using Incucyte^®^ ZOOM to Determine Levels of NET Formation in Subtypes of Adults and Children with Type 1 Diabetes

We first determined peripheral immune cell counts in the adult and pediatric cohorts of T1D and healthy donors. Despite slight variations in neutrophil counts, there was no difference between different subtypes of T1D patients compared to HC, both in adult and pediatric cohorts (Figure 3A). We also measured levels of peripheral markers of NET formation, such as plasma MPO and NE, in all donors. Plasma levels of MPO were not significantly different in the various subtypes of T1D donors in the adult and pediatric cohorts compared to HC donors (Figure 3B,C). NE levels were also comparable in the adult and pediatric cohorts (Figure 3D,E).

For a more direct and precise quantification of NET formation, we used Incucyte^®^ ZOOM to determine NET levels at baseline (unstimulated) and in response to selected concentrations of PMA (1 nM) and ionomycin (1 µM) in different subtypes of T1D patients and HC donors. The baseline levels of NETosis were under 3% in both adult and pediatric T1D and HC donors demonstrating that the isolation and staining protocols did not inadvertently activate the cells (Figure 4A). In the adult population, we observed slight variations, but no significant differences in the levels of NETosis at baseline and in response to either of the stimuli (Figure 4A–C, left and right panels, Appendix A). In the pediatric population, NET levels were higher, but not statistically significant, in donors with established T1D compared to HC donors at baseline and in response to ionomycin and PMA (Figure 4A–C, middle and right panels, Appendix A).

## 4. Discussion

The multifaceted characteristics and functional flexibility of neutrophils make them perfect candidates to foster immune dysregulation in autoimmune diseases, cancer, and chronic inflammatory conditions. Although dysregulated NETosis has been implicated in the pathophysiology of numerous autoimmune diseases, the lack of a standardized technique to objectively identify, analyze, and quantify the formation of NETing neutrophils has resulted in contradictory, often not significantly quantitative, data on their exact roles in these diseases.

In the case of T1D, most reports employ quantification of soluble NET remnants such as MPO, PR3, NE, and citrullinated histones (CitH3), as end-point surrogate markers of NETosis and do not objectively quantify the sequential dynamics and morphological changes of NETing neutrophils overtime [18,19,20,21,22]. Moreover, T1D is a heterogeneous disease in which younger age is associated with a higher risk and rate of progression through the developmental stages of the disease, which could lead to different NETosis patterns [23].

In the current study, we optimized the Incucyte^®^ ZOOM, a live-cell imaging platform, for an operator-independent, robust, and high-throughput quantification of peripheral blood-derived NETing neutrophils. After outlining a processing definition specific to identify, analyze, and quantify NETosis, we studied NET formation over a 6 h time course in HC donors in response to different doses of two NET-inducing stimuli known to employ distinct mechanisms. While PMA acts through the activation of NADPH and the release of granular proteins such as MPO and NE, ionomycin induces NET formation by the mobilization of calcium [10,28]. While the former was shown to produce a ‘suicidal’ form of NETosis, involving cell membrane rupture, calcium ionophores such as ionomycin mainly induce ‘vital’ NETosis, characterized by the externalization of chromatin and granular proteins through vesicular transport [29]. Although scarce, there are studies showing the existence of another form of NETosis involving the extrusion of mitochondrial DNA in a non-lytic manner [30,31,32]. Since all forms of NETosis involve the extravasation of DNA into the extracellular milieu, our live-cell imaging technique is applicable in all these cases [33]. It is, however, not possible to distinguish the different forms of NETosis, which remains a limitation of the technique. Therefore, the inclusion of additional dyes (i.e., Hoechst 33342™/MitoSox/MitoTracker) will be necessary [30]. Nevertheless, the Incucyte^®^ ZOOM platform allowed us to observe the integration of the two membrane-permeability-dependent DNA dyes into the NETing neutrophils overtime, thus revealing the kinetics of NETosis induced by each of the stimuli across multiple time points. The two stimuli incorporated the membrane-impermeable SYTOX^®^ green dye, which stains cells undergoing NETosis, in a dose-dependent chronologically and morphologically distinct manner, further confirming their mechanistic differences. While 1 nM of PMA induced a gradual incorporation of the dye approximately after 80 min of stimulation, neutrophils stimulated with 1 μM of ionomycin incorporated the dye rapidly but at a much later time point. Along with their characteristic curves showing NETosis during time, PMA and ionomycin showed distinct morphological characteristics as confirmed by high-resolution STED microscopy. Others also demonstrated that the Incucyte^®^ ZOOM platform can distinguish between different forms of cell death, including apoptosis, induced by various stimuli and inhibitors of NETosis [27]. The latter used this imaging technique to study NETosis in subtypes of neutrophils from people with SLE, the pathology of which can be partially attributed to neutrophil dysregulation.

To our knowledge, our team is the first to utilize this live-cell imaging platform to objectively identify, analyze, and quantify NETosis in both adults and children during different developmental stages of T1D in comparison to sex- and age-matched HC donors. Overall, people at different stages of T1D development, irrespective of age, had normal NET formation in real-time compared to HC. In the adult population, both basal and stimulated NET formation in all T1D groups were comparable to those in HC donors. Moreover, absolute neutrophil counts and peripheral levels of the soluble NET markers, MPO and NE, were comparable between the different groups. In contrast, others observed a reduction in circulating levels of PR3 and cell-free MPO-DNA complexes in established T1D patients [21,22]. However, in the former study, the difference in peripheral NET markers between the established T1D group and HC was lost after adjusting for absolute neutrophil counts. While MPO, NE, PR3, and MPO-DNA complexes are all considered biomarkers of NETosis, they have different protein affinities and structural characteristics, which might explain the observed discrepancies. In the pediatric population, we demonstrated that NET levels were comparable between all T1D and HC donors in response to ionomycin and PMA. Furthermore, soluble NET markers were similar between all donor groups. While others have not studied NETosis or NET remnants in pediatric patients with established T1D, a few studies found an increase in circulating levels of NE, PR3, MPO-DNA, and DNA remnants in children with newly diagnosed T1D [18,19,20]. Moreover, we did not observe significant differences in absolute neutrophil counts between the different groups of pediatric donors. In contrast, Wang et al. reported a (slight) reduction in neutrophil counts in pediatric T1D patients within one year of diagnosis. On the other hand, Kloperck et al. showed a significant increase in neutrophil counts in newly diagnosed T1D donors (within 7 days of diagnosis) compared to HC, at risk, and established T1D groups. The inconsistency of these data might be related to factors influencing peripheral absolute neutrophil counts, such as the general health of the donors at sample collection, as well as the influence of circadian variation on neutrophils [34].

When drawing conclusions on the role of NETosis in health and disease, it is important to distinguish between live imaging of NET formation in real-time and measuring circulating NET marker as end-point surrogates for NETosis. In the case of soluble NET markers, the results could be substantially influenced by interfering serum/plasma components and the need for assay standardization [35,36]. The presence of these soluble markers at the time of NETosis depends on the stimuli, as certain pathways of NET formation do not require the activation of MPO or NE [8,9,10,37]. Furthermore, NET markers have been shown to be released not only during NET formation, but also during degranulation in response to NET-inducing stimuli, which could bias the results when using these markers to quantify NETosis [38]. Of note, Wong et al. demonstrated an increase in NETosis in T1D patients compared to HC in response to ionomycin using manual counting of cells undergoing NETosis, which is highly subjective [39]. This further confirms the necessity of a robust and operator-independent technique for direct NET quantification overtime.

## 5. Conclusions

In summary, we described an imaging technique that allows for real-time, direct, and high-throughput analysis of NET formation, which results in robust and reliable data. We showed that this technique can be used to study NET formation in a heterogeneous disease such as T1D which demands subtyping of donors and an enhanced reproducibility of the results. Our data demonstrated that peripheral NET markers and NET formation are largely comparable between T1D subtypes and HC group, the latter determined by live-cell imaging. The use of the Incucyte^®^ ZOOM platform to study NETosis could significantly advance our understanding of the role of neutrophils in various diseases and potentially point towards a personalized approach in developing therapeutic strategies. Furthermore, our study highlights the importance of distinguishing between direct quantification of NETosis in real-time and measuring peripheral NET markers. We propose that combining different measures of NETosis could provide valuable insight into its role in health and disease.

## Figures and Tables

**Figure 1 biology-12-00882-f001:**
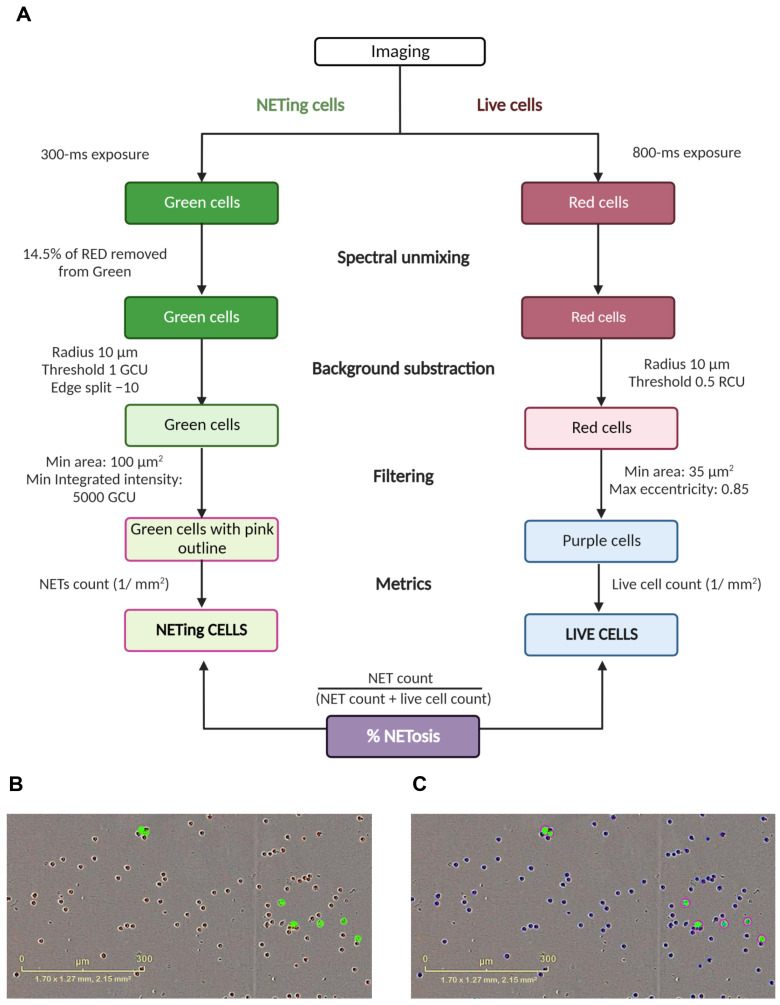
Incucyte^®^ ZOOM data analysis. (**A**) Stepwise schematic of the parameters of the processing definition employed to analyze the results of Incucyte^®^ ZOOM and determine percentage of NETosis in unstimulated and phorbol-myristate acetate (PMA)- and ionomycin-stimulated neutrophils. (**B**) Representative image of live neutrophils (red cells) and those undergoing NET formation (green cells) in response to 1 nM of PMA. (**C**) Analysis of the image in (**B**) using the parameters described in (**A**). Abbreviations: ms: millisecond; GCU: green calibrated unit; RCU: red calibrated unit. Scale bar = 300 µM.

**Figure 2 biology-12-00882-f002:**
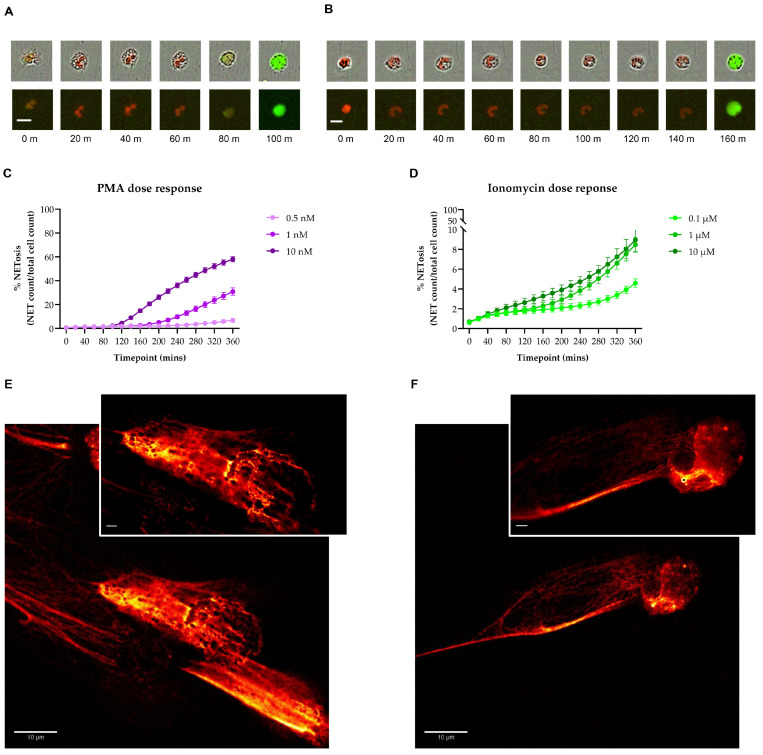
Induction of NETosis by distinct stimuli. (**A**,**B**) Comparison of morphological changes in neutrophils stimulated with phorbol-myristate acetate ((**A**), PMA, 1 nM) and ionomycin ((**B**), 1 µM). Single cells were imaged with combined red (NUCLEAR-ID^®^ red DNA stain) and green (SYTOX^®^ green dye) channels with (top panel) and without phase channel at 20 min (min: minutes) intervals using the 20× objective. Scale bar = 10 µM. (**C**,**D**) Dose response curves of neutrophils stimulated with increasing concentrations of PMA ((**C**), 0.5, 1, 10 nM) and ionomycin ((**D**), 0.1, 1, 10 µM) across multiple time points. (**E**,**F**) Representative images of PMA—((**E**), 1 nM) and ionomycin—((**F**), 1 µM) induced NETosis visualized by confocal (63× objective, scale bar = 10 µM) and stimulated emission depletion (STED) microscopy (top right corner inserts, 100× objective, scale bar = 2 µM).

**Figure 3 biology-12-00882-f003:**
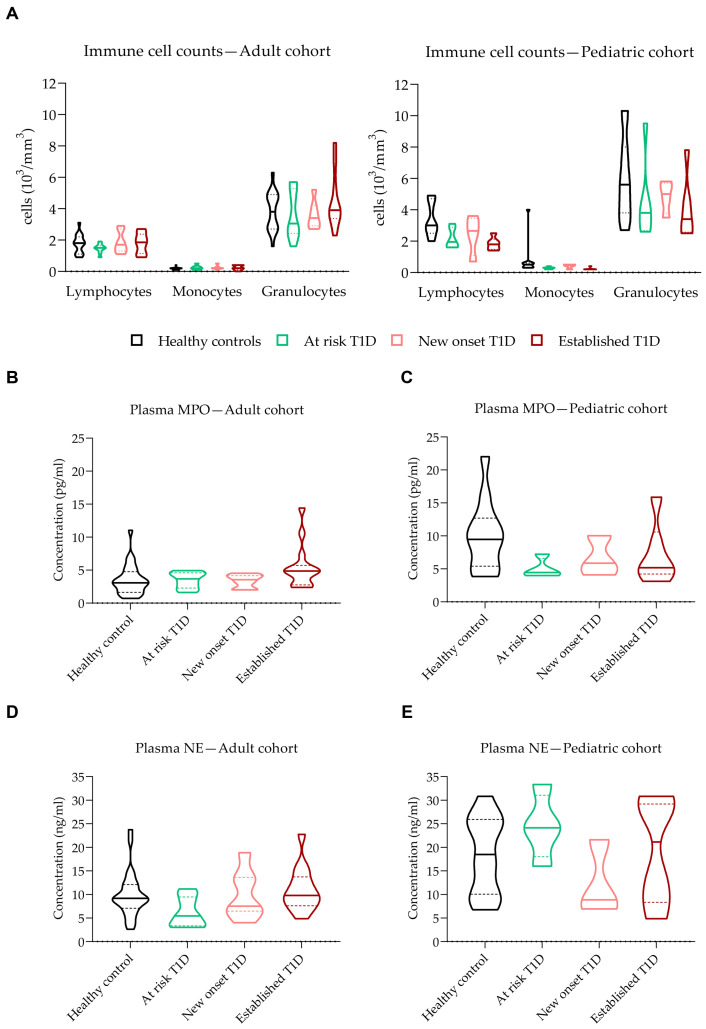
Peripheral markers of NET formation. (**A**) Lymphocyte, monocyte, and granulocyte counts (10^3^/mm^3^) in different subtypes of adult (left) and pediatric (right) T1D and HC donors. (**B**,**C**) Myeloperoxidase (MPO, pg/mL) and (**D**,**E**) neutrophil elastase (NE, ng/mL) levels in plasma of (**B**,**D**) adult and (**C**,**E**) pediatric T1D and HC subjects. Unpaired Kruskal–Wallis test with Bonferroni correction.

**Figure 4 biology-12-00882-f004:**
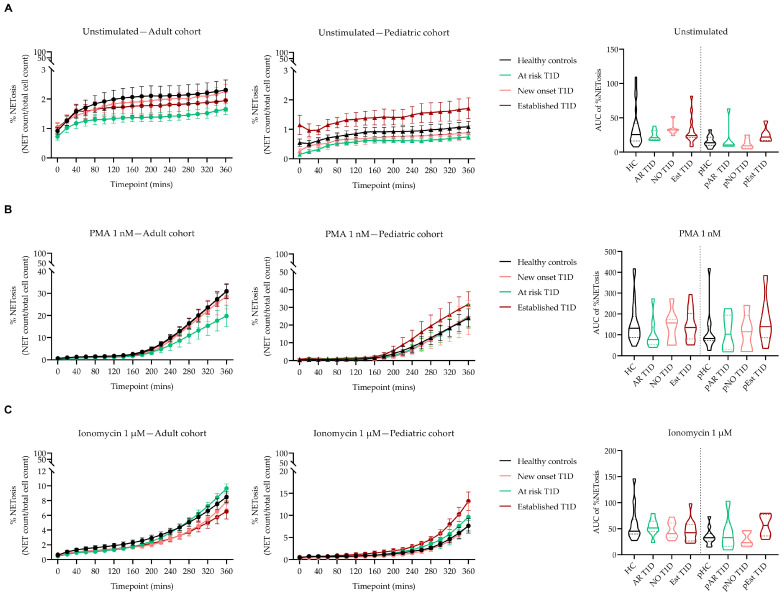
Using Incucyte^®^ ZOOM to determine NET levels in adult and pediatric T1D and HC donors. Graphs show levels of baseline ((**A**), unstimulated) and phorbol–myristate acetate—((**B**), PMA) and ionomycin—(**C**) stimulated NETosis in adult (left panel) and pediatric (middle panel) T1D and HC donors across multiple time points (20 min intervals). (**A**–**C**) Area under the curve (AUC) values of the graphs are represented (right panel). T1D donors are separated by subtype (new onset, at risk and established) according to the stage of disease development. Unpaired Kruskal–Wallis test with Bonferroni correction. Abbreviations: HC: healthy control; AR: at risk; NO: new onset; Est: established; pHC: pediatric healthy control; pAR: pediatric at risk; pNO: pediatric new onset; pEst: pediatric established.

**Table 1 biology-12-00882-t001:** Adult healthy control and patient donor characteristics.

	At Risk T1D	New Onset T1D	Established T1D	Healthy Control
Number of individuals	8	7	14	27
Age (years) ^1^	26 (21–33)	29 (22–42)	30 (20–45)	28 (22–45)
Disease duration (years) ^1^	NA	NA	11 (1–28)	NA
Disease duration (days) ^1^	NA	28 (13–49)	NA	NA
Age at diagnosis (years) ^1^	NA	29 (22–42)	19 (4–32)	NA
Gender F/M	4/4	5/2	6/8	17/10
Glycemia (mg/dL) ^1^	NA	105.4 (89–137)	176 (55–320)	NA
HbA1c (%) ^1^	NA	7.5 (6.5–10)	7.1 (5.0–9.2)	NA
TIR (%) ^1,2^	NA	NA	70 (37–90)	NA
Insulin dosage (U/day)	NA	24 (4–56)	49.0 (12–73)	NA
GADA (U/mL) pos/neg ^3^	4/4	6/1	8/6	0/27
IA2-A (U/mL) pos/neg ^4^	1/7	3/4	7/7	0/27
IAA (% binding) pos/neg ^5^	2/6	2/5	8/6	0/27
ZnT8A (% binding) pos/neg ^6^	1/7	0/7	1/13	0/27

^1^ mean (range), NA: not applicable. ^2^ time in range (TIR): time spent in between 70 and 180 mg/dL glycemia. ^3^ GADA titers (<23). ^4^ IA-2A titers (<1.4). ^5^ IAA titers (<0.6). ^6^ ZnT8A titers (<1.01).

**Table 2 biology-12-00882-t002:** Pediatric healthy control and patient donor characteristics.

	At Risk T1D	New Onset T1D	Established T1D	Healthy Control
Number of individuals	6	5	6	11
Age (years) ^1^	10 (8–14)	12 (6–16)	14 (9–17)	8 (5–13)
Disease duration (years) ^1^	NA	NA	7 (3–15)	NA
Disease duration (days) ^1^	NA	20 (7–42)	NA	NA
Age at diagnosis (years) ^1^	NA	12 (6–16)	8 (1–12)	NA
Gender F/M	3/3	3/2	3/3	5/6
Glycemia (mg/dL) ^1^	100 (66–125)	287 (90–740)	403.5 (213–502)	NA
HbA1c (%) ^1^	5.5 (5.3–5.7)	9.8 (7.3–13.4)	6.7 (6.5–7)	NA
TIR (%) ^1,2^	NA	NA	70.5 (69–72)	NA
Insulin dosage (U/day)	NA	32.2 (12–66)	48.8 (40–58)	NA
GADA (U/mL) pos/neg ^3^	5/1	3/2	3/3	0/11
IA2-A (U/mL) pos/neg ^4^	4/2	4/1	2/4	0/11
IAA (% binding) pos/neg ^5^	1/5	3/2	4/2	0/11
ZnT8A (% binding) pos/neg ^6^	1/5	3/2	1/5	0/11

^1^ mean (range), NA: not applicable, ^2^ time in range: time spent in between 70 and 180 mg/dL glycemia, ^3^ GADA titers (<23), ^4^ IA-2A titers (<1.4), ^5^ IAA titers (<0.6), ^6^ ZnT8A titers (<1.01).

## Data Availability

Not applicable.

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
