# Peer review of "High-Throughput Analysis of Neutrophil Extracellular Trap Levels in Subtypes of People with Type 1 Diabetes"

_biology, 2023, doi:10.3390/biology12060882_

Round 1

Reviewer 1 Report

The present manuscript highlights the Incucyte® ZOOM, a high-throughput live-cell imaging technique, as a robust and unbiased method to quantify NET formation, here, in this manuscript, in individuals at various stages of T1D development. The topic is very interesting, and the conclusion has been supported by experimental results. The most important results are: 1) PMA and ionomycin induced NETs with distinct kinetic characteristics; 2) in the paediatric population, NET levels were significantly higher in those with established T1D compared to the other subtypes and HC donors at baseline and in response to ionomycin, but not to PMA. Overall, the paper is well-written and scientifically sound. It is informative and interesting to read.

Although it is well described for the most part, there are some comments I believe need attention by authors prior to acceptance.

Some suggestions for revision are listed below:

It is important to clearly state what are the specifics of the method (Incucyte® ZOOM) and what are the specifics of NET, so that mistakes are not constantly repeated throughout further research.

For instance, in relation to the generalized schemes, there are differences in the presence/absence and the significance of molecules such as MPO (https://doi.org/10.1189/jlb.1211601, https://doi.org/10.7554/eLife.24437) for NETs formation. Therefore, MPO is not always component of NET; NET formation may occur in the absence of MPO, dependent on the stimulus (https://doi.org/10.1038/s41418-018-0261-x). In addition, as the authors themselves stated, plasmatic biomarkers such as MPO and NE have been shown to be released not only during NET and they are not specific for NET (methods that detect the presence of two markers are better, such as MPO-DNA, NE-DNA…).

IncuCyte ZOOM Imaging Platform permits the use of only two different fluorescent signals, it limits the ability to detect additional neutrophil components. Overall, this platform uses three imaging channels: the red fluorescent channel, the green fluorescent channel and the phase contrast channel. The authors adapted the protocol used by Gupta et al. 2018 (https://doi.org/10.4049/jimmunol.1700905). However, since this is a novel method, it is important to underline, for instance, that detection of cells that underwent NETs, was confirmed by visualization of the loss of nuclear lobulation, chromatin decondensation detected as an increase in nuclear diameter (100 μm2), and a decrease in the intensity of NUCLEAR-ID Red dye, followed by mixing of DNA content with cytoplasmic content and eventual cell membrane permeability and staining of nuclear content with SYTOX Green (Gupta et al. 2018). The details of the image processing steps and parameters of the processing definition are described only in Figure 2, without explanation. I think it is important to point this out, because the authors did not do an experiment like in Gupta 2018, where in an additional experiment they perform MPO staining (mouse anti-human MPO Ab FITC was added with the stimuli to the wells instead of SYTOX Green dye and imaged with phase contrast, red (1600-ms exposure), and green (800-ms exposure) channels) to confirm NETs.

Reviewer 2 Report

Please see comments for authors in the attached document.
